# AGENTICHYPOTHESIS: A SURVEY ON HYPOTHESIS GENERATION USING LLM SYSTEMS

**Adib Bazgir & Yuwen Zhang** *
Department of Mechanical and Aerospace Engineering
University of Missouri-Columbia
Columbia, MO 65211, USA
{abwbw,zhangyu}@missouri.edu

**Rama chandra Praneeth Madugula**
Department of Mechanical Engineering
New York University
New York, NY 10012
{rm6057}@nyu.edu

## ABSTRACT

Hypothesis generation is a cornerstone of scientific discovery, enabling researchers to formulate testable ideas that drive innovation and progress. Although traditional methods have proven effective, they are often constrained by cognitive biases, information overload, and time limitations. Recently, the emergence of Large Language Models (LLMs) has introduced a promising approach to automate and enhance hypothesis generation. Trained on extensive datasets, these models have demonstrated the potential to generate novel, testable hypotheses across various domains. This review critically examines state-of-the-art methodologies for LLM-based hypothesis generation, including Retrieval Augmented Generation (RAG), multi-agent frameworks, and iterative refinement techniques. Applications in biomedical research, materials science, product innovation, and interdisciplinary studies are discussed, highlighting both the versatility and the impact of these systems. Despite their promise, LLMs also present challenges such as hallucinations, data biases, and ethical concerns, which necessitate careful implementation and oversight. Future research directions include refining model architectures, integrating multimodal capabilities, and establishing robust ethical frameworks to optimize the use of LLMs in scientific research. Overall, this review provides a balanced overview of how LLMs may enhance hypothesis generation while also addressing the associated challenges. The GitHub repository containing open-source LLM-empowered hypothesis generation systems is available at https://github.com/adibgpt/AgenticHypothesis.

## 1 INTRODUCTION

Hypothesis generation is fundamental to scientific discovery and technological innovation, providing a systematic framework for proposing and testing novel ideas that drive advancement across diverse fields. These include biomedicine, psychology, materials science, NLP, astronomy, machine learning, automated program repair (APR), ecology, linguistics, molecular biology, and social sciences. Historically, hypothesis generation relied on human intuition, manual literature reviews, and heuristic-based systems. However, these approaches increasingly fall short in addressing modern challenges such as information overload, cognitive biases, and the growing complexity of interdisciplinary research. Key barriers to traditional methods include the inefficiency of manual processes, reliance on predefined rules, and limited scalability. For example, in biomedicine, the overwhelming volume of literature obscures critical therapeutic insights, while in astronomy and neuroscience, traditional methods fail to unify insights from disparate datasets, leaving knowledge gaps unaddressed. Similarly, APR struggles with dynamic software bugs, and social sciences face challenges in integrating structured and unstructured data. These limitations underscore the need for scalable, automated tools that enhance hypothesis generation by systematically exploring the expansive search space of potential solutions. The advent of Large Language Models (LLMs) has introduced a paradigm shift in hypothesis generation, addressing these limitations by leveraging extensive pretraining on diverse datasets. LLMs dynamically synthesize information, identify latent patterns,

---

*Corresponding Author: Yuwen Zhang at zhangyu@missouri.edu.

and propose innovative, testable ideas across disciplines, democratizing access to hypothesis-driven exploration and accelerating discovery.

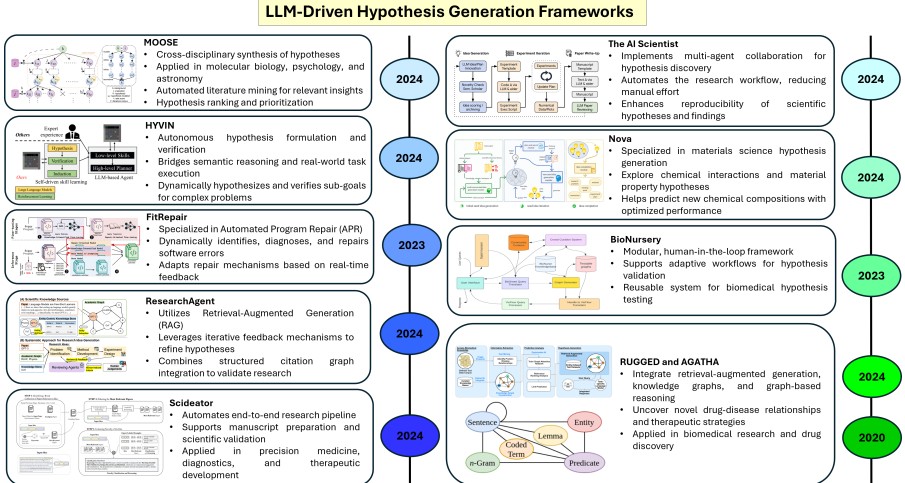

Figure 1: Featured LLM-driven hypothesis generation frameworks.

LLMs have emerged as transformative tools for hypothesis generation, integrating advanced reasoning techniques, modular workflows, and structured methodologies to overcome traditional limitations. Models like GPT-4 (Park et al., 2023), BioGPT (Luo et al., 2022), SciBERTT (Beltagy et al., 2019), and PMC-LLaMA (Wu et al., 2023), and frameworks like SciHypo (Tadiparthi et al., 2024), MOOSE (Yang et al., 2024a), HYVIN (Peng et al., 2024), FitRepair (Xia et al., 2023), ResearchAgent (Baek et al., 2024), and The AI Scientist (Lu et al., 2024), exemplify how LLMs enhance the novelty, diversity, and efficiency of hypothesis generation as further disclosed in Figure 1.

SciHypo (Tadiparthi et al., 2024) and MOOSE (Yang et al., 2024a) enable cross-disciplinary synthesis and hypothesis generation across fields such as molecular biology, psychology, and astronomy. Meanwhile, HYVIN (Peng et al., 2024) autonomously hypothesizes and verifies sub goals in reinforcement learning environments, effectively bridging semantic reasoning with real world task execution. In automated program repair, FitRepair (Xia et al., 2023) applies the "plastic surgery hypothesis" to dynamically address software bugs, and ResearchAgent (Baek et al., 2024) leverages retrieval augmented generation (RAG) along with iterative feedback to produce scientifically rigorous hypotheses. Additionally, the AI Scientist (Lu et al., 2024) integrates multi agent collaboration to automate the research workflow and enhance reproducibility. In biomedical research, LLMs generate hypotheses for drug discovery, therapeutic mechanisms, gene expression patterns, and even surface passivation strategies for perovskite solar cells. In materials science, tools like MOOSE-Chem and Nova enhance chemical hypothesis generation by exploring complex interactions in material properties. Within machine learning and ecology, LLMs uncover novel diffusion models and adaptive ecological mechanisms by leveraging interdisciplinary insights. In the realm of automated program repair (APR), systems such as FitRepair (Xia et al., 2023) dynamically identify and resolve software bugs. For linguistics and psychology, LLMs simulate human-like generalization behaviors to deepen our understanding of language acquisition and improve causal graph integration. Finally, in the social sciences, LLMs connect hypotheses with supporting evidence from the literature, thereby facilitating robust hypothesis evidencing and synthesis. LLMs employ citation graph integration to ground hypotheses in existing literature, thereby enhancing scientific rigor while pushing the boundaries of innovation. Additionally, techniques such as chain of thought reasoning and retrieval-augmented generation facilitate iterative reasoning and refinement, ensuring that the hypotheses generated are both rigorous and diverse. Despite their transformative potential, LLMs face challenges such as hallucinations, reliance on pre existing data, computational inefficiencies, and occasional implausible outputs. Addressing these issues requires human oversight, multi agent collaboration, and iterative feedback mechanisms to ensure reliability and scalability.

This unified review synthesizes advancements in LLM driven hypothesis generation, spanning methodologies, applications, challenges, and complementary strategies across diverse scientific do-

mains. It explores frameworks such as SciHypo (Tadiparthi et al., 2024), MOOSE (Yang et al., 2024a), HYVIN (Peng et al., 2024), FitRepair (Xia et al., 2023), ResearchAgent (Baek et al., 2024), and The AI Scientist (Lu et al., 2024), which utilize techniques like causal graph analysis, iterative planning, and retrieval augmented generation to address various research needs. The review highlights the transformative potential of LLMs in domains including biomedical research, automated program repair (APR), materials science, machine learning, psychology, and the social sciences. It also confronts critical challenges such as data bias, computational inefficiencies, hallucinations, error mitigation, and ethical considerations, while advocating for complementary strategies like multi agent collaboration, transparency focused designs, and hybrid neuro symbolic approaches to improve reproducibility, reliability, and overall innovation.

## 2 METHODOLOGIES FOR HYPOTHESIS GENERATION USING LLMS

Synthesizing insights from a comprehensive review of methodologies reported in the literature, this review highlights the transformative potential and versatility of Large Language Models (LLMs) in hypothesis generation. The reviewed studies span a wide range of domains including biomedical research, social sciences, molecular biology, computational biology, linguistics, chemistry, big data, consumer product ideation, and web based data exploration and collectively underscore how LLMs are reshaping scientific inquiry. A common theme across these studies is the emphasis on modular workflows, iterative refinement, and domain specific adaptations, which are further enhanced by multi modal integration and advanced reasoning tools. Together, these approaches, as schematically provided in Figure 2, facilitate the generation of hypotheses that are scientifically robust, innovative, actionable, and contextually relevant, thereby marking a significant advancement in hypothesis driven research.

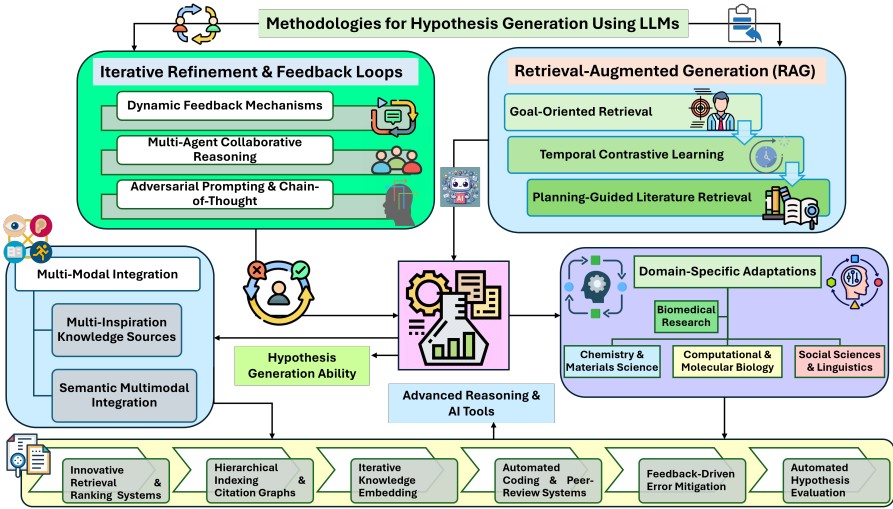

Figure 2: Utilized methodologies for LLM-driven hypothesis generation approaches.

### 2.1 CORE APPROACHES

A central theme across these methodologies is the use of iterative refinement frameworks in which hypotheses are generated, validated, and dynamically improved through structured workflows. For instance, dynamic feedback mechanisms exemplified by frameworks like Nova employ self-correction, retrieved knowledge, and multi-agent feedback loops to iteratively refine hypotheses (Tong et al., 2024; Peng et al., 2024; Hu et al., 2024; Qi et al., 2024). Moreover, collaborative reasoning systems simulate brainstorming by assigning specialized roles such as ideation, evaluation, and critique, effectively mirroring real-world scientific processes (Peng et al., 2024; Park et al., 2024) (Park et al., 2023). Additionally, techniques like adversarial prompting and chain of thought prompting facilitate inductive reasoning, enabling dynamic hypothesis validation and continuous iterative refinement (Ciucă et al., 2023; Jamil et al., 2023; Lu et al., 2024).

RAG based frameworks leverage large datasets and pre trained knowledge to enhance the contextual relevance and quality of hypotheses. They employ goal oriented retrieval and temporal contrastive learning to dynamically identify knowledge gaps and model evolving relationships within temporal knowledge graphs, thereby uncovering novel associations (Zhou et al., 2024a; Hu et al., 2024). In addition, planning-guided retrieval offers a structured approach to retrieving and ranking relevant literature, effectively grounding and validating the generated hypotheses (Jamil et al., 2023; Skarlinski et al., 2024).

Multi modal frameworks synthesize diverse data types, including textual, visual, and structured data, to create contextually rich hypotheses. They draw on multi inspiration knowledge sources by combining molecular interactions, experimental results, patents, and more to identify impactful relationships (Yang et al., 2024b; Hu et al., 2024). Additionally, semantic multi modal integration merges data across various domains to support interdisciplinary hypothesis generation (Zhang et al., 2018; Misra & Kim, 2024).

## 2.2 DOMAIN-SPECIFIC ADAPTATIONS

LLM driven frameworks leverage domain specific models like BioBERT, knowledge graphs, and curated datasets such as PubMed to generate clinically relevant hypotheses. Temporal partitioning and graph-based reasoning ensure the novelty and accuracy of hypotheses addressing challenges like drug discovery and therapeutic pathways (Qi et al., 2023; Tadiparthi et al., 2024; Mitta, 2023; Sybrandt et al., 2020; Zhou et al., 2024a; Qi et al., 2024). Multi inspiration frameworks synthesize knowledge from diverse chemical and material properties, employing evolutionary algorithms to uncover novel phenomena (Yang et al., 2024b; Hu et al., 2024). Frameworks replicate workflows from high impact publications, combining RNA sequencing analyses with computational modeling to explore genetic correlations with diseases. Knowledge ecosystems enable mapping of natural language queries into structured graphs for hypothesis testing (Xia et al., 2024; Misra & Kim, 2024; Bersenev et al., 2024). Customized datasets and annotation frameworks incorporate linguistic and behavioral cues to address tasks like deception detection, stress analysis, and language acquisition. Feature-specific experiments simulate human like learning processes to generate hypotheses about syntactic generalization patterns (Wang et al., 2024; Ishikawa, 2024; Koneru et al., 2024; Misra & Kim, 2024). Domain specific constraints such as demographics and pricing guide hypothesis generation in consumer product ideation, ensuring relevance and feasibility. Lightweight web crawling systems efficiently collect and process niche domain data for hypothesis exploration in areas like regulatory compliance (wag, 2024; Lin et al., 2024; Li et al., 2024b).

## 2.3 INNOVATIVE TOOLS

Dynamic retrieval strategies employ hierarchical indexing and citation traversal to identify and rank literature for effective hypothesis grounding (Skarlinski et al., 2024; Yang et al., 2024b). Additionally, iterative knowledge embedding leverages temporal and graph-based embeddings to model evolving knowledge relationships, further enriching the hypothesis generation process (Yang et al., 2024b; Zhou et al., 2024a). Error mitigation systems employ feedback-driven loops that iteratively refine hypotheses to ensure robustness and alignment with scientific standards (Bersenev et al., 2024; Lu et al., 2024). In addition, peer review emulation is integrated into these systems, where automated evaluations assess hypotheses based on criteria such as novelty, soundness, and relevance (Lu et al., 2024). To foster interdisciplinary innovation, Facet Recombination Models propose combining complementary facets of a concept, such as its purpose, underlying mechanism, and evaluation methods (Radensky et al., 2025). This approach is enhanced by incorporating Statistical and Bayesian Reasoning, utilizing probabilistic programming to validate generated hypotheses and ensuring a balance between creative exploration and practical feasibility (Qi et al., 2024).

## 2.4 ADVANCEMENTS IN ALGORITHMS

To further enhance the hypothesis generation and refinement process, Swarm Based and Dialog Driven Models simulate peer-review by incorporating a multitude of perspectives (Park et al., 2024; Li et al., 2024a). Complementing this, Ensemble Methods bolster hypothesis robustness through the integration of diverse computational insights and domain specific knowledge (Park et al., 2023; Lu et al., 2024). Hypotheses can be iteratively improved using Dynamic Validation Systems, which

employ LLM powered reviewers and feedback mechanisms for continuous refinement (Bersenev et al., 2024; Baek et al., 2024). Furthermore, Chain of Ideas Models contextualize these hypotheses within evolving research trends, enabling advancements across dynamic fields of study (Li et al., 2024a).

## 3 APPLICATIONS ACROSS SCIENTIFIC DOMAINS

The widespread adoption of large language models (LLMs) and AI driven hypothesis generation frameworks has profoundly influenced scientific research across multiple domains. These advancements have not only accelerated hypothesis generation but also introduced novel methodologies for refining, validating, and implementing scientific insights. In biomedical research, LLMs and multi agent systems have enhanced drug discovery, disease modeling, and genomic analysis, offering data driven solutions for therapeutic innovation. Similarly, materials science has benefited from AI powered frameworks that facilitate the exploration of novel materials, synthesis techniques, and performance optimization strategies. The structured nature of AI enhanced ideation has also proven valuable in product development, streamlining innovation cycles in industries such as pharmaceuticals, biotechnology, and advanced manufacturing. Furthermore, these methodologies have fostered cross-disciplinary collaborations, enabling knowledge integration across biomedical, computational, environmental, and social sciences. By bridging gaps between traditionally distinct fields, AI driven frameworks are redefining research paradigms, promoting efficiency, and driving innovation across diverse scientific landscapes.

### 3.1 BIOMEDICAL RESEARCH

The integration of large language models (LLMs) and advanced computational frameworks has fundamentally transformed biomedical research, enabling the automated generation, refinement, and validation of hypotheses across diverse applications (Chai et al., 2024; Yang et al., 2024a). These advancements leverage iterative workflows, domain specific fine tuning, and retrieval augmented reasoning to address pressing challenges such as drug discovery, molecular interactions, and genomic analysis (Abdel-Rehim et al., 2024; Qi et al., 2024). Multi agent systems and graph based methodologies have been particularly impactful in accelerating the identification of novel drug disease relationships and therapeutic strategies, allowing for the synthesis of previously unlinked biomedical insights (Pelletier et al., 2024; Sybrandt et al., 2020). Further, modular and human in the loop frameworks have demonstrated the ability to facilitate adaptive hypothesis testing, enhancing the efficiency and reliability of generated hypotheses (Jamil et al., 2023; Peng et al., 2024). Such methodologies prioritize explainability, enabling researchers to systematically refine computationally derived insights through interactive and iterative validation mechanisms (Xia et al., 2023; 2024). Additionally, the incorporation of temporal modeling and AI driven literature synthesis has proven instrumental in capturing time evolving biomedical relationships, allowing for the prediction of emerging disease pathways and the identification of novel molecular mechanisms (Zhou et al., 2024a; Park et al., 2023). These advancements, particularly when coupled with automation tools that streamline the entire research pipeline from hypothesis generation to manuscript preparation significantly accelerate progress in precision medicine, diagnostics, and therapeutic innovation (Radensky et al., 2025; Lu et al., 2024).

### 3.2 MATERIALS SCIENCE AND ENGINEERING

Hypothesis driven research in materials science has similarly benefited from AI powered frameworks, facilitating the systematic exploration of material properties, synthesis techniques, and performance optimization strategies (Qi et al., 2023; Wang et al., 2024). The ability of LLMs to generate hypotheses regarding advanced alloys, solid electrolytes, and nanomaterials has enabled rapid advancements in energy efficient and high-performance materials (Zhou et al., 2024b; Zaitsev et al., 2023; Liu et al., 2024). By integrating computational simulations with experimental data, researchers have been able to systematically model and optimize material behaviors under various conditions, thereby bridging the gap between theoretical predictions and practical applications (Yang et al., 2024b; Farinhas et al., 2023). Structured approaches that iteratively refine hypotheses based on multi source data synthesis have emerged as a powerful strategy for materials innovation (Spangler et al., 2014; Sybrandt et al., 2020). Techniques incorporating graph based reasoning and

semantic embeddings have been particularly useful in identifying previously unrecognized material property relationships, leading to breakthroughs in the design of sustainable and energy efficient materials (Koneru et al., 2024; Yang et al., 2024b). High throughput automated discovery pipelines further accelerate material innovation, reducing the reliance on purely experimental trial and error methodologies (Radensky et al., 2025; Park et al., 2023). These advancements highlight the transformative role of AI driven hypothesis generation in materials science, facilitating the development of next generation composites, nanotechnology applications, and renewable energy materials (Lu et al., 2024).

### 3.3 INNOVATION AND PRODUCT DEVELOPMENT

The structured and automated methodologies that enhance biomedical and materials science research have also significantly influenced innovation and product development processes (Qi et al., 2024; Ciucă et al., 2023). AI driven hypothesis generation frameworks enable the rapid ideation, validation, and refinement of novel concepts, reducing time-to-market for innovative solutions (Takagi et al., 2023; il Lee et al., 2024). Structured brainstorming methodologies and adversarial prompting strategies have demonstrated the ability to generate high quality, diverse hypotheses that align with market needs (Demartini et al., 2024; Pelletier et al., 2024). By leveraging automated refinement techniques, industries such as biotechnology, pharmaceuticals, and advanced manufacturing can systematically evaluate and prioritize high-potential innovations (Radensky et al., 2025; Zhang et al., 2018). The integration of probabilistic reasoning and full cycle automation facilitates robust hypothesis testing, enabling more efficient decision making processes (Hu et al., 2024; Si et al., 2024). Additionally, scalable AI driven frameworks allow for the exploration of unconventional yet scientifically grounded product ideas, fostering innovation across various sectors (Tang et al., 2023; Chen et al., 2024). These methodologies not only streamline ideation but also ensure that product concepts are scientifically validated and aligned with evolving industry demands (Lu et al., 2024).

### 3.4 CROSS-DISCIPLINARY CONTRIBUTIONS

The increasing adaptability of AI driven hypothesis generation frameworks has fostered interdisciplinary research, facilitating collaboration across a wide array of scientific fields (Ishikawa, 2024; O'Brien et al., 2024). By integrating knowledge across biomedical, computational, and environmental sciences, AI powered methodologies have enabled the synthesis of novel insights that address complex global challenges (Li et al., 2024c; Qi et al., 2024). The ability to model time evolving relationships across disciplines has proven especially valuable in areas such as climate science, public health, and computational biology, where dynamic interactions between multiple variables necessitate robust, data-driven analyses (Mitta, 2023; Farinhas et al., 2023). The modular, explainable, and scalable nature of AI driven frameworks has also played a critical role in supporting interdisciplinary hypothesis generation (Proebsting & Poliak, 2024; Lu et al., 2024). The incorporation of structured validation pipelines ensures that cross domain insights remain relevant and actionable, allowing for the development of innovative solutions that transcend traditional disciplinary boundaries (Skarlinski et al., 2024). As a result, these advancements have driven progress in areas such as computational sciences, environmental modeling, and public policy, where collaborative, multi domain research is essential for addressing emerging scientific and societal challenges (Lu et al., 2024).

## 4 EVALUATION METRICS AND CHALLENGES

Evaluating AI generated hypotheses requires a rigorous framework to ensure innovation, reliability, and applicability. According to Figure 3, this section explores key evaluation criteria, implementation challenges, and ethical concerns in hypothesis generation. The first subsection, Metrics for Assessing Hypotheses, highlights essential factors like novelty, accuracy, feasibility, efficiency, and diversity, alongside specialized dimensions such as scalability, impact, reproducibility, and scientific rigor. These metrics ensure hypotheses are innovative, evidence based, and practical for real world application. However, Challenges in Implementation persist, including computational demands, data dependencies, hallucinations, and alignment issues, which impact the reliability and scalability of AI driven research. Lastly, Ethical and Practical Concerns emphasize the need for transparency, bias mitigation, scientific integrity, and equitable access to ensure responsible AI deployment. Address-

ing these challenges will enhance the trustworthiness, inclusivity, and effectiveness of AI generated hypotheses in advancing scientific discovery.

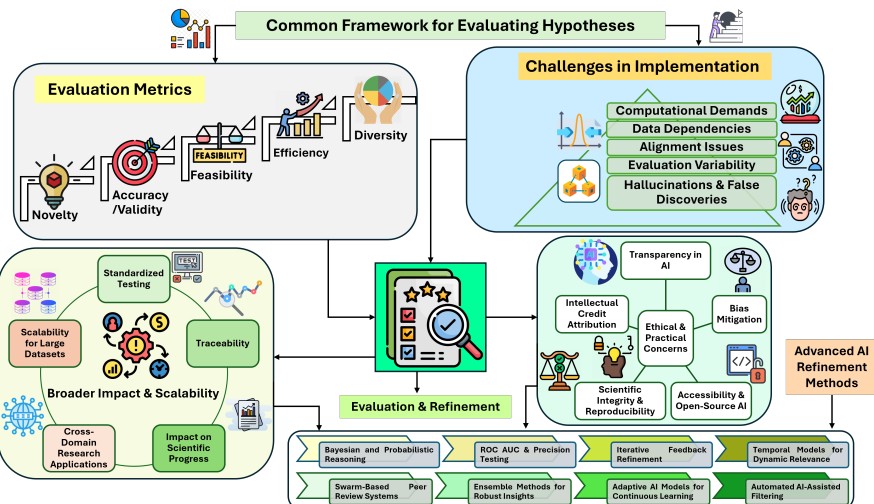

Figure 3: The schematic of framework workflow used for evaluating hypotheses.

## 4.1 METRICS FOR ASSESSING HYPOTHESES

Novelty is consistently emphasized as the cornerstone of hypothesis evaluation, driving innovation through uncharted ideas in areas such as novel drug combinations in biomedical research (Abdel-Rehim et al., 2024; Mitta, 2023), mechanisms in chemistry (Yang et al., 2024b), or interdisciplinary links (O'Brien et al., 2024). Methodologies like iterative refinement, facet recombination, feedback loops, and semantic filtering ensure systematic exploration of unique hypothesis spaces (Tong et al., 2024; Hu et al., 2024; Lu et al., 2024). Accuracy/validity ensures alignment with empirical data, logical reasoning, and benchmarks, validated through precision metrics like ROC AUC (Spangler et al., 2014) and experimental testing in fields like drug synergy (Abdel-Rehim et al., 2024) or astrophysics (Zaitsev et al., 2023)). Feasibility evaluates the practicality of hypotheses, ensuring they are actionable within technological and resource constraints, as seen in diagnostic frameworks (Batista & Ross, 2024) or material science (Chen et al., 2024). Efficiency measures resource optimization and workflow streamlining, with examples such as BioNursery's scalability (Jamil et al., 2023) or lightweight configurations for linked data processing (wag, 2024). Diversity ensures a wide-ranging exploration of hypothesis spaces, avoiding redundancy and fostering creativity, particularly in frameworks promoting exploration across domains (Xiong et al., 2024; Zhou et al., 2024a)

Relevance aligns hypotheses with user defined goals or domain-specific challenges, employing tailored prompts and contextual refinement (Demartini et al., 2024; Skarlinski et al., 2024). Scalability measures frameworks' ability to handle large datasets or integrate multi modal data, such as distributed embeddings in biomedical research (Sybrandt et al., 2020). Impact assesses the broader significance of hypotheses in advancing scientific understanding, such as replicating high impact studies or identifying novel correlations in biomedical contexts (Bersenev et al., 2024; Qi et al., 2024). Reproducibility ensures consistent outputs through standardized workflows and rigorous experimental validations (Cudré-Mauroux et al., 2012; Ifargan et al., 2024). Traceability strengthens transparency by linking hypotheses to their data sources and computational processes (Ifargan et al., 2024; Park et al., 2023). Quality integrates multiple evaluation metrics to uphold high standards in hypothesis generation and testing (Li et al., 2024a; Lu et al., 2024). Temporal relevance is crucial in dynamic fields, ensuring hypotheses reflect evolving trends and knowledge (Zhou et al., 2024a). Scientific rigor evaluates the methodological robustness and precision of hypotheses, ensuring alignment with established principles and standards (Bersenev et al., 2024; Park et al., 2023).

## 4.2 CHALLENGES IN IMPLEMENTATION

Computational demands remain significant due to resource-intensive workflows involving transformer models, iterative refinement, and large-scale embeddings (Qi et al., 2024; Yang et al., 2024b; Bersenev et al., 2024). Data dependencies constrain hypothesis quality and novelty, as frameworks rely heavily on training datasets, limiting exploration beyond known domains (Park et al., 2023; Meincke et al., 2024). Alignment issues arise in reconciling AI-generated hypotheses with external systems or domain-specific requirements, such as semantic coherence in temporal graphs (Zhou et al., 2024a) or contextual accuracy in biomedical terminologies 2023. Evaluation variability reflects inconsistencies between human and automated reviews, particularly for subjective metrics like novelty or impact (Skarlinski et al., 2024; Qi et al., 2024). Hallucinations, where plausible but unfounded hypotheses emerge, persist in fields like biomedical research, mitigated through retrieval augmented generation and adversarial prompting (Xiong et al., 2024; Pelletier et al., 2024). Iterative inefficiencies occur as novelty and originality plateau during repeated refinement cycles, necessitating balanced optimization (Radensky et al., 2025; Lu et al., 2024). User adoption barriers stem from the complexity of multi-stage frameworks or computational intensity, limiting accessibility for non-expert users 2023, 10.1145/3584371.3613022. Semantic reasoning limitations and representation variability further challenge nuanced causal or temporal hypothesis generation in scientific contexts (Braun & Kunz, 2024; Koneru et al., 2024)

## 4.3 ETHICAL AND PRACTICAL CONCERNS

Transparency is promoted through detailed workflows, explainable AI tools, and open-source frameworks, although the complexity of systems like AGATHA or BioNursery can hinder interpretability for general users (Jamil et al., 2023; Lu et al., 2024). Bias mitigation ensures fair representation across domains, addressing imbalances in training data to support equitable exploration of ideas in underrepresented fields (Proebsting & Poliak, 2024; Meincke et al., 2024). Scientific integrity underscores rigorous validation and reproducibility to avoid disseminating flawed or unreliable hypotheses, especially in high-stakes areas like biomedical research (Batista & Ross, 2024; Li et al., 2024b). Accessibility remains a critical challenge due to computational demands, with lightweight models, modular workflows, and open-source tools proposed to democratize access (Xiong et al., 2024; Demartini et al., 2024; Park et al., 2023). Intellectual credit attribution emphasizes recognizing contributions from AI systems, human collaborators, and data curators in generating hypotheses (Ifargan et al., 2024). Responsibility in AI applications highlights safeguarding against unethical or unsafe practices, ensuring compliance with ethical standards in deployment (Li et al., 2024b; Lu et al., 2024). Equity promotes fair distribution of resources and balanced representation in AI-driven research, particularly in domains like financial inclusion and biomedical innovation (Sharma et al., 2024; Chen et al., 2024).

## 5 FUTURE DIRECTIONS

The rapid evolution of large language models (LLMs) in hypothesis generation has demonstrated their transformative potential across diverse scientific domains. However, further advancements are required to enhance their effectiveness, address ethical concerns, and expand their applicability. This section outlines key future directions, including technological innovations, ethical frameworks, and the broadening of LLM applications.

## 5.1 TECHNOLOGICAL INNOVATIONS

To fully leverage the capabilities of LLMs in scientific hypothesis generation, future developments must focus on enhancing grounding mechanisms, contextual understanding, and multimodal learning. Enhanced LLM grounding requires that hypotheses be well supported by empirical evidence and domain-specific knowledge; thus, future advancements should integrate deeper retrieval augmented generation (RAG) techniques, knowledge graph embeddings, and dynamic fact-checking models to improve the alignment of generated hypotheses with verified scientific literature and datasets. Additionally, improved contextual understanding is essential, as LLMs often struggle with capturing nuanced, domain-specific knowledge; fine-tuning models with specialized corpora, employing reinforcement learning from human feedback (RLHF), and implementing structured reason-

ing frameworks can enhance contextual comprehension, while integrating chain-of-thought prompting and discourse aware generation will refine the ability to maintain logical consistency across extended scientific narratives. Finally, the future of hypothesis generation lies in multimodal learning the fusion of text, images, graphs, and structured data where multimodal LLMs capable of processing diverse data formats, such as experimental datasets, molecular structures, and astronomical observations, will enable richer, interdisciplinary hypothesis generation through techniques like vision-language models, structured data embeddings, and cross domain transfer learning.

## 5.2 ETHICAL FRAMEWORKS

As LLMs become integral to scientific research, establishing ethical guidelines and transparency measures is paramount to ensure responsible and equitable deployment. AI generated hypotheses should be interpretable and traceable to their data sources; thus, implementing robust documentation standards, citation tracking mechanisms, and audit trails is essential to improve accountability in AI driven research. Additionally, because LLMs can inherit biases from their training data, future systems should incorporate fairness aware training methodologies, debiasing algorithms, and diversity-enhancing pretraining strategies, along with standardized transparent reporting on model limitations and bias assessments. Ensuring the reliability of AI generated hypotheses requires rigorous validation protocols, which can be achieved by implementing hybrid human AI peer review systems, embedding uncertainty quantification mechanisms, and fostering open source verification initiatives to uphold scientific standards. Moreover, to democratize access to these advanced tools, LLM driven hypothesis generation should be accessible across diverse research communities, including those in resource-constrained settings, through open source AI frameworks, decentralized model hosting, and community-driven AI governance structures.

## 5.3 EXPANDING APPLICATIONS

The application of LLMs in hypothesis generation has predominantly focused on well-established scientific fields, but future research should explore underrepresented domains and interdisciplinary integration. LLMs hold potential in underexplored areas such as archaeology, environmental science, behavioral economics, and emergent materials discovery; expanding training datasets to include historical texts, ecological patterns, and alternative scientific paradigms can facilitate hypothesis generation in these fields. Additionally, bridging insights across traditionally distinct disciplines can lead to groundbreaking discoveries LLMs should be optimized for synthesizing knowledge across fields like computational neuroscience, bioinformatics, and social ecological systems, with semantic embeddings that link diverse scientific terminologies to enhance cross-domain hypothesis formation. Moreover, AI driven research should not be confined to elite institutions; cloud-based platforms, federated learning models, and AI powered collaborative research networks can empower independent researchers, citizen scientists, and students worldwide, while incorporating multilingual AI systems will promote knowledge dissemination across linguistic barriers.

## 6 CONCLUSION

The transformative potential of Large Language Models (LLMs) in hypothesis generation is reshaping the landscape of scientific discovery. By leveraging structured methodologies, retrieval-augmented generation, multi-agent collaboration, and iterative refinement, LLMs have proven to be valuable tools in diverse fields ranging from biomedical research to social sciences. Despite these advancements, challenges such as data biases, computational inefficiencies, and ethical concerns must be addressed to ensure responsible deployment and widespread accessibility. Moving forward, enhancing LLM grounding, multimodal learning, and contextual reasoning will be crucial for improving their reliability and applicability. Ethical frameworks emphasizing transparency, fairness, and scientific integrity must be established to maintain trust in AI-generated research. Additionally, expanding applications into underrepresented domains and fostering interdisciplinary collaborations will further unlock the potential of LLMs in accelerating innovation. Ultimately, the future of AI-driven hypothesis generation lies in a synergistic approach where technological advancements, ethical considerations, and broad accessibility converge to create a more robust and equitable research ecosystem. By integrating these principles, LLMs can serve as powerful catalysts for scientific progress, democratizing knowledge and shaping the future of inquiry across disciplines.

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

# A  APPENDIX

## A.1  COMPARATIVE ANALYSIS

In this section, a comprehensive comparison of the current AI-driven hypothesis generation frameworks is provided, with the aim of elucidating both their strengths and inherent limitations. The analysis is structured into three subsections. First, the strengths and weaknesses of existing systems are examined, with features such as structured reasoning, iterative refinement, and modular workflows being highlighted for their contribution to effectiveness, while challenges such as reliance on curated datasets and computational inefficiencies are noted. Next, emerging trends and patterns that are shaping the evolution of these frameworks are identified, ranging from the integration of hybrid models and enhanced explainability to the shift toward end-to-end automation. Finally, critical research gaps are discussed, including the need for more versatile data integration, standardized evaluation metrics, and improved scalability and error mitigation. A clear snapshot of the current landscape is provided by this comparative analysis, and insights into potential avenues for future innovation in AI-driven hypothesis generation are offered.

### A.1.1  STRENGTHS AND WEAKNESSES OF CURRENT SYSTEMS

The frameworks reviewed demonstrate significant advancements in hypothesis generation, highlighting key features such as structured reasoning, domain-specific adaptations, iterative refinement, modular workflows, and end-to-end automation. Systems like SciMON (Chai et al., 2024) and FieldSHIFT (O'Brien et al., 2024) leverage domain-specific datasets and interdisciplinary insights, enabling novel and highly relevant hypotheses tailored to their respective fields. Similarly, HypoGeniC (Zhou et al., 2024b), SciHypo (Tadiparthi et al., 2024), and MOOSE-Chem (Yang et al., 2024b; Qi et al., 2024) highlight structured approaches and iterative refinement to improve logical consistency, novelty, and domain relevance. Frameworks such as MOOSE (Yang et al., 2024a), RUGGED (Pelletier et al., 2024), and KnIT (Spangler et al., 2014) demonstrate the utility of modular workflows, enhancing scalability and adaptability. Tools like AGATHA (Sybrandt et al., 2020), GAIA (Zhang et al., 2018), and BioNursery (Jamil et al., 2023) excel in integrating graph based reasoning and multimodal data, enabling effective hypothesis validation. Systems like Chain-of-Ideas (CoI) (Li et al., 2024a) and Nova (Hu et al., 2024) focus on iterative planning, ranking hypotheses, and ensuring novelty through self correcting feedback loops. Advanced systems such as the AI Scientist (Lu et al., 2024)Sources and related content and Multi-Agent LLM (Qi et al., 2024) expand these methodologies to automate entire research pipelines, integrating tools like PubMed, coding assistants, and multidimensional evaluation metrics to enhance feasibility, novelty, and reproducibility. However, limitations remain. Many systems depend heavily on curated datasets or structured inputs, such as PubMed (Abdel-Rehim et al., 2024; Tadiparthi et al., 2024), E-CARE datasets (Tang et al., 2023), or knowledge graphs (Jamil et al., 2023; Zhou et al., 2024a; Baek et al., 2024). This reliance restricts adaptability to dynamic, unstructured, or real-time data sources. Computational inefficiencies, particularly in iterative workflows and multi-agent systems, hinder scalability. Moreover, while novelty is a consistent strength, feasibility and robustness often require human intervention, reducing autonomy.

### A.1.2  EMERGING TRENDS AND PATTERNS

Recent advancements in AI-driven hypothesis generation have showcased a diverse array of methodologies that integrate structured reasoning and graph-based approaches, iterative refinement, domain-specific adaptations, hybrid models, end-to-end automation, and enhanced explainability. Many systems now leverage structured knowledge sources such as causal graphs (Tong et al., 2024; Xiong et al., 2024), knowledge-grounded frameworks (Jamil et al., 2023), and graph reasoning (Sybrandt et al., 2020; Zhou et al., 2024a) to boost scalability, relevance, and the validity of generated hypotheses. Frameworks like HypoGeniC (Zhou et al., 2024b), BRAINSTORM (Tang et al., 2023), and Nova (Hu et al., 2024) employ iterative refinement techniques—using adversarial prompting, contrastive learning, and feedback loops—to continuously improve the quality and robustness of their hypotheses. Recognizing the importance of tailoring methodologies to specific challenges, systems such as SciMON (Chai et al., 2024), SciHypo (Tadiparthi et al., 2024), and MOOSE-Chem (Yang et al., 2024b) (Yang et al., 2024b) adapt their approaches to meet domain-specific needs, underscoring the value of domain expertise in guiding AI research. Additionally,

tools like GAIA (Zhang et al., 2018), SHi (Koneru et al., 2024), and Multi-Agent LLM (Qi et al., 2024) illustrate the potential of hybrid models by integrating structured data with language models, thereby combining the strengths of structured and unstructured sources. Furthermore, frameworks such as AI Scientist (Lu et al., 2024) and Multi-Agent LLM (Qi et al., 2024) demonstrate the trend toward end-to-end automation, streamlining entire research workflows and reducing the need for human intervention. Finally, a focus on explainability and evaluation transparency is evident in systems like FieldSHIFT (O'Brien et al., 2024), ML-integrated frameworks (Batista & Ross, 2024), and CoI (Li et al., 2024a), which emphasize the necessity for standardized benchmarks that assess novelty, interpretability, and reproducibility.

### A.1.3 RESEARCH GAPS

Recent research in AI-driven hypothesis generation highlights several key challenges and opportunities that must be addressed to further advance the field. Many current frameworks rely heavily on curated datasets or structured inputs, which limits their applicability in less organized, multi-modal, or real time contexts. Expanding systems such as BioNursery (Jamil et al., 2023), GAIA (Zhang et al., 2018), and ResearchAgent (Baek et al., 2024) to integrate diverse data types would significantly enhance their versatility. Additionally, while these systems consistently produce novel hypotheses, practical feasibility often lags behind; incorporating domain-specific constraints and real time validation mechanisms—as demonstrated in frameworks like HypoGeniC (Zhou et al., 2024b), NovaHu et al. (2024), and GPT-4( (Park et al., 2023) could help balance this trade-off. Moreover, the iterative workflows and multi-agent systems employed by platforms like AI Scientist (Lu et al., 2024) and MOOSE (Yang et al., 2024a) introduce significant resource demands, making the optimization of computational processes essential for improving scalability and accessibility. Another critical area is the need for standardized evaluation metrics, as many frameworks currently lack unified benchmarks to comprehensively assess hypothesis quality across dimensions such as interdisciplinary relevance, reproducibility, and fairness. In parallel, issues related to error reduction and robustness evidenced by instances of hallucinations, fabricated experimental details, and biased outputs in systems like AI Scientist (Lu et al., 2024) and GPT-4 (Park et al., 2023) underscore the necessity for improved error-checking and validation mechanisms. Ethical and transparency considerations also play a crucial role, particularly given the potential misuse of AI systems in generating fraudulent research or overwhelming peer review processes, which calls for the establishment of robust ethical guidelines and transparent deployment protocols.Finally, although frameworks like MOOSE-Chem (Yang et al., 2024b) (Yang et al., 2024b) and KnIT (Spangler et al., 2014) are capable of generating high quality hypotheses, their limited integration with experimental workflows and real time validation systems indicates that bridging this gap is essential for creating more effective end-to-end research pipelines.

