# OpenReview forum: "AgenticHypothesis: A Survey on Hypothesis Generation Using LLM Systems"
_ICLR.cc/2025/Workshop/AgenticAI — ICLR 2025 Workshop AgenticAI Poster_

### Official Review · Reviewer_RaVo · 2025-02-27
**Hypothesis Generation with Large Language Models: Opportunities and Challenges**

**Rating:** 5
**Confidence:** 5

**Review:**

# Summary:
This survey provides a comprehensive overview of current approaches utilizing LLMs for hypothesis generation. The authors present a taxonomy of existing approaches and analyze their advantages, limitations, and potential future directions.

# Strength:
1. The survey is well-structured, making it easy to follow.
2. The future directions highlight promising research avenues.

# Weakness:
1. The review of existing methods can be more detailed rather than just listing them. E.g. , in section 4.2 about hallucinations, a deeper discussion on how this issue has been addressed, its bottleneck, and potential solutions would be beneficial.
2. A summary of commonly used datasets in this area would enhance the survey's completeness and usability.
3. Section 4.1 only mentions metrics but does not provide insights into the current SOTA performance. Including a comparison of top-performing methods would strengthen the survey’s analysis.

---

### Official Review · Reviewer_cGG8 · 2025-03-02
**Review of AgenticHypothesis**

**Rating:** 5
**Confidence:** 5

**Review:**

Summary:
This survey examines LLM-based hypothesis generation, highlighting methods like Retrieval-Augmented Generation (RAG) and multi-agent frameworks. It also discusses applications in various domains such as biomedical research, materials science, etc. While LLMs enhance discovery, challenges like biases and hallucinations persist. Future directions include refining architectures, integrating multimodal data, and strengthening ethical safeguards for responsible implementation.

Strengths:
1) This paper provides a comprehensive review of LLM-based hypothesis generation, covering key methodologies like Retrieval-Augmented Generation (RAG), multi-agent frameworks, and iterative refinement techniques.
2) The review does not just highlight the benefits of LLMs but also shows their limitations, such as hallucinations, biases, and ethical concerns, providing a well-rounded analysis.
3) By identifying future research directions, the paper offers valuable guidance for advancing LLM-based hypothesis generation.

Weaknesses:
1) The paper primarily summarizes existing methods but lacks case studies or deeper insights into the effectiveness of different LLM-based hypothesis generation approaches.
2) Although the paper mentions challenges like hallucinations and biases, it does not dive deeply into mitigation strategies or real-world case studies.

---

### Official Review · Reviewer_4hMs · 2025-03-03
**AgenticHypothesis: A Survey on Hypothesis Generation Using LLM Systems**

**Rating:** 6
**Confidence:** 3

**Review:**

Paper Summary:

The paper hypothesizes that LLM-based systems can overcome the limitations of traditional hypothesis-generation methods by leveraging advanced reasoning techniques, modular workflows, and structured methodologies to generate novel, scientifically rigorous hypotheses across diverse fields.

Strengths:
- Structured Methodological Analysis: The survey effectively categorizes different methodological approaches (iterative refinement, RAG-based, multi-modal frameworks), making a complex field more accessible.

- Multi-Domain Application Review: The paper effectively showcases how LLM-based hypothesis generation works across diverse scientific fields.

- Balanced Evaluation Framework: The survey offers a multidimensional framework for evaluating hypothesis quality, providing comprehensive assessment criteria.

Weaknesses:
- The paper acknowledges hallucination challenges but fails to explore mitigation strategies across different systems sufficiently.

- The review overemphasizes fully automated systems while neglecting hybrid approaches where scientists and LLMs collaborate to generate hypotheses.

- The paper lacks quantitative comparisons between LLM-based hypothesis systems, hindering objective assessment of their relative effectiveness.

Questions to Authors:

 See above.

---

### Decision · Program_Chairs · 2025-03-05

Accept (Poster)